# Gradual Enhancement of the Assemblage Stability of the Reed Rhizosphere Microbiome with Recovery Time

**DOI:** 10.3390/microorganisms10050937

**Published:** 2022-04-29

**Authors:** Fuchao Zheng, Xiaoming Mou, Jinghua Zhang, Tiange Zhang, Lu Xia, Shenglai Yin, Lingye Wu, Xin Leng, Shuqing An, Dehua Zhao

**Affiliations:** 1Institute of Wetland Ecology, School of Life Science, Nanjing University, Nanjing 210023, China; fczheng2018@163.com (F.Z.); xm_mou@163.com (X.M.); mg1830019@smail.nju.edu.cn (J.Z.); ztg9626@163.com (T.Z.); lulu8668@yeah.net (L.X.); lengx@nju.edu.cn (X.L.); 2College of Life Sciences, Nanjing Normal University, Nanjing 210023, China; shenglai.yin@outlook.com; 3Changshu Wetland Conservation and Management Station, Changshu 215500, China; lyeahwu@163.com; 4Nanjing University Ecology Research Institute of Changshu, Changshu 215500, China

**Keywords:** reeds, rhizosphere microbiome, ecological restoration, co-occurrence network, assemblage stability

## Abstract

Rhizoplane microbes are considered proxies for evaluating the assemblage stability of the rhizosphere in wetland ecosystems due to their roles in plant growth and ecosystem health. However, our knowledge of how microbial assemblage stability is promoted in the reed rhizosphere of wetlands undergoing recovery is limited. We investigated the assemblage stability, diversity, abundance, co-occurrence patterns, and functional characteristics of reed rhizosphere microbes in restored wetlands. The results indicated that assemblage stability significantly increased with recovery time and that the microbial assemblages were capable of resisting seasonal fluctuations after more than 20 years of restoration. The number of bacterial indicators was greater in the restoration groups with longer restoration periods. Most bacterial indicators appeared in the 30-year restoration group. However, the core taxa and keystone species of module 2 exhibited greater abundance within longer recovery periods and were well organized, with rich and diverse functions that enhanced microbial assemblage stability. Our study provides insight into the connection between the rhizosphere microbiome and recovery period and presents a useful theoretical basis for the empirical management of wetland ecosystems.

## 1. Introduction

Ecological restoration measures implemented by plants and microbes serve as natural efforts to promote the robustness of ecosystems and create better habitats for terrestrial organisms [1,2,3]. Microbial multifunctionality can increase the stability of an ecosystem, especially with respect to rhizosphere microbes with high diversity and abundance, which are noted by ecologists as a research hotspot [4,5]. However, the relationship between restorative efficacy and microbial assemblage stability in terms of ecosystem homeostasis remains poorly understood. Therefore, it is necessary to clarify the recovery effects on microbial assemblage stability by demonstrating the characteristics of the microbiome in restored wetlands, especially the rhizosphere microbiota, due to its high diversity and numerous functions [6].

Long-term restoration effectively shapes microbial structure and improves diversity due to a stable soil environment in terrestrial ecosystems [7,8]. Generally, with a longer duration of restoration, greater microbial diversity and richness result in many functions, microbial assemblages become more stable [4,9,10] and the stable assemblage with multifunctionality helps strengthen the resilience of the ecosystem [9,11,12]. However, the microbial assemblages inhabiting different niches encounter biotic and abiotic perturbations, which may disturb the stability of the assemblages [13,14]. A previous study indicated that bacterial assemblages with higher phylogenetic diversity tended to be more stable, implying that microbiomes with higher biodiversity are more resistant to perturbation [15]. Several studies have revealed that the perturbations associated with climate change, such as rainfall and high temperature, affect microbial assemblages and suggested that microbial assemblages with higher diversity have the ability to resist disturbance, while such disturbance barely influences the resistance and resilience of key soil microbial functional groups [16,17]. Nevertheless, there are still very few studies illustrating the dynamics of the diversity and assemblage stability of the rhizosphere microbiome in a series of restored wetlands.

Network analysis is a popular approach for obtaining new perspectives on microbial assemblages. In addition to providing information about richness and composition, the network approach adds a substantial dimension to our understanding of microbial community ecology [18,19,20]. For example, recent studies that utilized differential abundance or network analysis revealed that several microbes or microbiota served as indicators of the health of soil, coral reefs, and seagrass [21,22,23]. Mycorrhizal fungi can effectively improve plant diversity, accelerate succession, and promote the re-establishment of missing plants in restored communities [3]. In addition, most of these organisms serve as an essential driver of nutrient metabolism, suggesting that they play major roles in host health and ecosystem homeostasis [24,25]. Moreover, numerous studies indicated that the core taxa of the microbiome played a vital role in maintaining the stability of the microbial assemblages, and the core taxa typically dominated in terms of abundance [26]. The core taxa inhabiting different corals are capable of resisting seasonal fluctuations because of their mutualism [27]. Within a cluster of a cooccurrence network, the core taxa have similar ecological preferences and make major contributions to functional stability, especially that represented by a positive relationship between carbon metabolism and functional stability [9]. Previous studies indicated that keystone species with many connections in the co-occurrence network drove community composition and function irrespective of their abundance [19,28,29], and the key function of keystone species acted in maintaining the stability of community structure [30]. For example, keystone taxa such as *Nitrospira* and *Gemmatimonas* perform nitrogen metabolism and phosphonate and phosphinate metabolism to enhance assemblage stability in the soil environment [15]. However, to date, very few studies have focused on the core taxa and keystone species of the rhizosphere microbiome and few studies have revealed their key roles in maintaining the stability of microbial assemblages.

*Phragmites australis* (Cav.) Trin. ex Steud. (reed), perennial and cosmopolitan macrophytes that are widely distributed [31], are often used as model plants. Reeds can form relationships with microbes and play a role in phytoremediation to contribute to ecosystem functions that are displayed by determining the processes, dynamics and cycling of carbon, nitrogen, phosphorus, nutrients and water in aquatic or terrestrial ecosystems [32,33,34]. Previous studies have documented that the lineage-specific control of rhizosphere bacteria and within-lineage bacterial communities was similar among reed communities [35]. Most studies involving reeds have focused on invading effects [36,37], decomposition efficiency [38,39], and single kingdom or functional groups [32,40]. To the best of our knowledge, the recovery effects on the assemblage stability of the reed rhizosphere microbiome and the vital roles of core taxa and keystone species in maintaining assemblage stability remain unexplored.

With these perspectives, we investigated the impact of intervals of a decade of restoration on reed rhizosphere microbial assemblages during the level, wet and dry seasons using amplicon sequencing and network analysis. We aimed to address the following questions: (1) How does the recovery period affect reed rhizosphere microbial diversity and assemblage stability? (2) Which microbes serve as indicators of the stability of microbial assemblages? (3) What are the abundances of the core taxa and keystone species and their roles in improving the stability of assemblage in the reed rhizosphere?

## 2. Materials and Methods

### 2.1. Location Selection and Sampling

We selected reeds as the research materials, all of which were octoploid (8×) plants from Lake Taihu Basin [31]. All sampling locations were in four restored wetlands (Appendix A), where projects returning farmlands or fishing ponds back to wetlands were carried out in the 1980s, 1990s, 2000s, and 2010s. However, we found very few suitable sample locations with recent restoration or without restoration nearby. A total of 14 transects were established in the four restored wetlands in the level, wet and dry seasons of 2020. Four samples were collected from each transect (each transect measured 30 m, and four sampling locations were established along each transect (*n* = 4 samples per transect)), and a total of 168 samples were collected. The arrangement of sampling locations along each transect is illustrated in Appendix A. In general, respective three transects were sampled in the Shanghu Wetland that has been restored for 40 years (40 y, 120.6853 E, 31.6537 N), the Shajiabang Wetland that has been restored for 30 years (30 y, 120.8021 E, 31.5548 N) and the Nanhu Wetland that has been restored for 10 years (10 y, 120.6307 E, 31.5927 N); five transects were sampled in the Taihu Wetland that has been restored for 20 years (20 y, 120.3596 E, 31.3238 N) (Appendix A). The coordinates of the transects and basic information are listed in Appendix A. We selected a 15 cm × 15 cm section of reeds at each location. The reeds were dug out to obtain the rhizosphere soil samples, which were placed into sample bags. The soil samples were stored on solid carbon dioxide (dry ice). Rhizosphere soil samples were stored at −80 °C for DNA extraction.

### 2.2. DNA Extraction and High-Throughput Sequencing

Total DNA was extracted by following the Cetyltrimethylammonium bromide (CTAB) method (see Appendix A for more detail). The extracted DNA was evaluated on a 1.0% agarose gel, and the DNA concentration and purity were determined with a NanoDrop 2000 UV–vis spectrophotometer (Thermo Scientific, Wilmington, NC, USA). The 16S rRNA/ITS genes of distinct regions (16S V4 and ITS1) were amplified using specific primers 515F (GTGCCAGCMGCCGCGGTAA)806R (GGACTACHVGGGTWTCTAAT) and ITS1F (CTTGGTCATTTAGAGGAAGTAA)-ITS2 (GCTGCGTT CTTCATCGATGC) [41]. An appropriate amount of the prepared sample DNA was placed in a centrifuge tube, and then the sample was diluted to 1.0 ng/μL with sterile water. The diluted genomic DNA was used as a template according to the selection of the sequencing region with specific primers with barcodes. New England Biolabs’ Phusion^®^ High-Fidelity PCR Master Mix with GC buffer (New England Biolabs, Ipswich, MA, USA) and high-efficiency and high-fidelity enzymes for PCR to ensure amplification efficiency and accuracy. Library sequencing was performed using the Ion Plus Fragment Library Kit 48 rxns (Thermo Scientific, Waltham, MA, USA) following the manufacturer’s recommendations. The library quality was assessed on a Qubit@ 2.0 fluorometer (Thermo Scientific, Waltham, MA, USA). Finally, the library was sequenced on an Ion S5TM XL platform, and 400 bp/600 bp single-end reads were generated.

### 2.3. Data Analysis

#### 2.3.1. Microbial Diversity and Stability Index Calculation

We used alpha diversity (whole-tree phylogenetic diversity (PD)) to analyze the complexity of assemblage diversity. PD was calculated with QIIME (Version 1.7.0) and visualized with R software, and one-way analysis of variance (ANOVA) was used to test the difference in in the PD index with recovery time. We calculated the variation degree based on the abundance of OTUs per sample. The average variation degree (AVD) is typically calculated to estimate the temporal stability of microbial assemblages, and the assemblage stability of the reed rhizosphere microbiome was evaluated based on the AVD [15]. The taxa with a relative abundance > 0.001% served as predictors of the microbial assemblage stability index [9], which was calculated and visualized using R software. Linear regression analysis between alpha diversity and recovery periods in different seasons was performed, and we analyzed the relationships between the PD index and the microbial assemblage stability index with quadratic regression. All analyses were conducted by R software (Version 3.6.1).

#### 2.3.2. Bipartite Network and Microbial Indicators

Microbial assemblage composition analysis was utilized in the “vegan” package of R software. We determined the top 30 phyla based on the relative abundance of the bacterial assemblages and all the fungal phyla to analyze the composition of microbes in the reed rhizosphere. Analysis of Similarities (ANOSIM) was performed between the groups [42]. Correlation based on indicator species analysis was conducted with the R package “indicspecies” [43]. Differential OTU abundance was tested between one or more of the restorative periods of the reed rhizosphere (OTU tables with the same threshold) of both kingdoms using likelihood ratio tests with the R package “edgeR” [44]. The OTU table of bacterial and fungal indicators (hereafter BIs and FIs) was obtained. We visualized the significant (*p* < 0.05) operational taxonomic unit (OTU) associations with one or more of the different recovery periods from the indicator species analysis using bipartite networks [45]. Quadratic regression analysis between the assemblage stability index and accumulated abundance of microbial indicators was visualized in R software (Version 3.6.1).

#### 2.3.3. Co-Occurrence Network and Functional Prediction

Core taxa were defined as those confirmed by both indicator species analysis and likelihood ratio tests [45]. Co-occurrence networks of the reed rhizosphere microbiome were constructed, and the “trimmed means of M” were utilized to normalize counts per million. We conducted Spearman rank correlations between species and visualized the positive, significant correlations (R > 0.7 and *p* < 0.001). We calculated topological network properties, including the number of nodes, number of edges and degrees of cooccurrence, by using the “igraph” package [46]. The nodes represent species, the edges represent positive and significant correlations between species, and the degrees of co-occurrence represent direct correlations to a node. The greedy optimization of the modularity algorithm was utilized to discern network modules, which were substructures of nodes with a higher density of edges within groups than between them [47]. We identified core taxa of modules (hereafter: M) in the whole network. The keystone species were determined for the reed rhizosphere meta-network and defined as those nodes with the top 1.0% of node degree values for the whole network [45]. We used ANOVA to examine the different abundances of core taxa and keystone species with recovery time and visualized them with boxplots. Quadratic regression was used to analyze the relationships between the microbial assemblage stability and the accumulated abundance of core taxa and keystone species in the modules. The PICRUSt 2 (Phylogenetic Investigation of Communities by Reconstruction of Unobserved States) tool based on the eggNOG (evolutionary genealogy of genes: Nonsupervised Orthologous Groups) database was used for the prediction of bacterial function, and the FUNGuild database was used to annotate the fungal function [48,49,50]. We presented the top 10 in relative abundances of functional profiles in the bacterial assembly module, and heatmaps were generated for the differentiating modules that used the “pheatmap” package.

## 3. Results

### 3.1. Alpha Diversity and the Assemblage Stability

A total of 13,340,997 16S rRNA genes and 13,453,338 ITS sequences were selected for classification from reed rhizosphere soil, and 12,428 and 6970 OTUs were obtained from bacteria and fungi, respectively. Alpha diversity and the assemblage stability index were used to explain the status of rhizosphere microbes with the effects of recovery time. The results revealed that bacterial PD indices were significantly different with recovery time, which showed a significant increase with increasing recovery periods in the level (R^2^ = 0.07, *p* = 0.03) and dry seasons (R^2^ = 0.17, *p* < 0.01), whereas the fungal PD indices significantly differed with recovery time in the level and wet seasons and obviously increased in the wet season (R^2^ = 0.11, *p* < 0.01) (Appendix A; Figure 1a–c). The values varied within a relatively small range between bacteria and fungi, and the values of fungi were higher than those of bacteria in the wet season (Figure 1b). However, the assemblage stability index showed a significant increase with an increase in the recovery period (*p* = 0.016), but the value in the 30 y restoration period was slightly higher than that in the 20 y restoration period (Figure 2a and Appendix A), and the number of OTUs was 6283, 5939, 5694 and 6860 in the 40 y, 30 y, 20 y and 10 y recovery periods, respectively (Appendix A). The results indicated that the assemblage stability index significantly differed in the 10 y restoration period (*p* = 0.001), but there was no significant difference in the restoration periods longer than 20 years (Figure 2b–e and Appendix A). The relationship between the assemblage stability index and PD index was significant (R^2^ = 0.598; *p* = 0.016), and the fitted value approached the maximum in the 30 y restoration period, while the fungal PD index exhibited a positive relationship with the assemblage stability index (R^2^ = 0.225) (Figure 1d,e).

### 3.2. Bacterial and Fungal Indicators

The number of species was counted from the reed rhizosphere among the four periods of restored wetlands according to the taxonomic identification results. There were 4127 and 1681 genera affiliated with 71 and 14 bacterial and fungal phyla, respectively, and the top 30 phyla and genera based on relative abundance are shown in Appendix A. The bacterial and fungal assemblages exhibited similar phyla, but the proportion of the relative abundance varied among the different seasons. For example, in the bacterial assemblages, Proteobacteria, Cyanobacteria, Bacteroidetes, Acidobacteria, Nitrospirae, Chloroflexi, Euryarchaeota, and Actinobacteria were the dominant bacterial phyla, and the fungal phyla Ascomycota, Basidiomycota, Rozellomycota, GSOl, and Monoblepharomycota were the dominant taxa. However, dissimilarity tests showed that bacterial and fungal assemblages were significantly different between any two groups at recovery time (Appendix A). Then, we used all the OTUs to select the bacterial and fungal indicators (hereafter BIs and FIs) whose abundances significantly differed (*p* < 0.05) with recovery time. The number of BIs was 6 times higher than that of FIs. Specifically, 1839, 2319, 1557 and 1104 BIs and 233, 448, 163 and 324 FIs occurred in the four restored wetlands (Figure 3a,b and Appendix A). With longer restoration times, the BIs gradually increased, while the proportion of unique species declined (Figure 3a and Appendix A). Surprisingly, the BIs in the 40 y group and the percentages of unique species in the 10 y group were lower than those in the 20 y and 30 y groups in the wet season (Appendix A). However, the FIs represented more than 80% of the species individually residing in the four restored wetlands, and all of the FIs in the 10 y or 20 y group were unique species in the wet or dry season (Appendix A). Among the three seasons, the indicators exhibited great variations in the BIs, Proteobacteria, Cyanobacteria, Firmicutes and Bacteroidetes were the most abundant taxa in the reed rhizosphere (Figure 3a), whereas Ascomycota was the most abundant fungal assemblage (Figure 3b). The results revealed that the BIs presented a negative relationship with the assemblage stability index (R^2^ = 0.192), which was positively associated with the FIs (R^2^ = 0.264) (Figure 3c,d). Thus, the bipartite networks showed that the restoration period of the BIs was closely related to the relatively long restoration periods and that the BIs increased during sustained restoration, thus serving as ideal organisms for inferring the stability of the microbial assemblages. In contrast, the FIs had a large number of unique species during restoration, and they did not enhance assemblage stability.

### 3.3. Modules of Core Taxa and Functional Prediction

There were 2394 (1510 bacteria and 884 fungi) nodes in the co-occurrence networks, and the connections between and within bacteria and fungi are shown in Figure 4a. The network width was 5.31, and most connections appeared in the bacterial assemblages. However, the core taxa were found in the MON, and we discovered that they agglomerated by module. We selected the top four in the relative abundance of core taxa of modules in the MON (Figure 4a). The results indicated that the core taxa differed among recovery times. For example, in M2 (342 bacteria and 0 fungi), relatively high abundance and strong connections of core taxa were observed in the 30 y and 40 y restoration periods, while in M6 (84 bacteria and 6 fungi), M1 (104 bacteria and 90 fungi) and M8 (63 bacteria and 6 fungi), the highest abundance of core taxa occurred in the 30 y, 20 y and 10 y restoration periods, respectively (Figure 4c–f; Appendix A). The most abundant core bacterial taxa were Proteobacteria, Cyanobacteria, Bacteroidetes, Firmicutes and Acidobacteria, and the core taxa of fungi were Ascomycota, Basidiomycota, Rozellomycota and Glomeromycota (Figure 4b). However, there were 22 bacterial keystone species and no fungal keystone species in the MON, and the abundance of keystone species significantly increased with increasing recovery times (*p* = 0.006) (Figure 5e). The bacterial keystone species belonged to Proteobacteria, Nitrospirae, Acidobacteria, Chloroflexi, Spirochaetes, Bacteroidetes, Latescibacteria, and Euryarchaeota. (Appendix A). Moreover, the relationships between assemblage stability and the relative abundances of core taxa and keystone species were analyzed in this study, and the results suggested that the keystone species helped promote the stability of the microbial assemblages (R^2^ = 0.451) (Figure 5a). In addition, the core taxa of M2 displayed a higher abundance in the relatively long periods, which enhanced assemblage stability (R^2^ = 0.361) (Figure 5b). However, the core taxa of M8 with high abundance in the 10 y restoration period did not improve assemblage stability (Figure 5c).

Then, we used the combined OTUs in the modules to predict functions based on the eggNOG and FUNGuild databases. The top 10 relative abundances of functional profiles with bacterial core taxa and keystone species are presented in Figure 6. For example, the main functions performed by bacteria included glycosyltransferase involved in cell wall biosynthesis (COG0438,0463), nucleoside-diphosphate-sugar epimerase (COG0451), DNA-binding transcriptional regulator, and the LysR family (COG0583). The results indicated that the majority of bacterial OTUs were assigned to 14 functional features, which were predominantly involved in carbohydrate transport and metabolism, cell motility, signal transduction mechanisms, cell wall/membrane/envelope biogenesis, and signal transduction mechanisms (Appendix A). Likewise, the functions of fungal core taxa included plant pathogens, fungal parasites, plant pathogen soil saprotrophs, wood saprotrophs, leaf saprotrophs, soil saprotrophs, arbuscular mycorrhizae, and ectomycorrhizae. The fungal OTUs were assigned to eight trophic mode groups, e.g., pathotrophs, pathotroph–saprotrophs, pathotroph–saprotroph–symbiotrophs, and symbiotrophs (Appendix A). Moreover, the prevalence of functional profiles was similar for different modules, while the abundance of functional profiles was different among the modules (Figure 6).

## 4. Discussion

### 4.1. Long-Term Restoration Enhances Microbial Assemblage Stability

Several studies have indicated that ecological restoration has positive effects on the development of microbial assemblages [51,52]. For example, previous studies indicated that reforestation served as an effective ecological restoration strategy to increase soil microbial diversity and help promote the assemblage stability of the soil microbiome [7,9]. The current results revealed that the longer the restoration period was, the greater the bacterial diversity and assemblage stability in the reed rhizosphere of the restored wetlands (Figure 1 and Figure 2). Because reeds are perennial, the reed community generally increases over the recovery time, and thus the microbial assemblage stability increases with recovery time, and the greater assemblages possess a stronger ecological resilience [53]. The current study revealed that the PD index fluctuated with seasonal change, and the values were similar between bacterial and fungal assemblages in the level and dry seasons, while the fungal values were higher than the bacterial values in the wet season (Figure 1b). This result is consistent with that in a previous report [54]. A previous study indicated that phylogenetic diversity presented a positive relationship with the community stability of the soil microbiome [15]. In this study, the microbial assemblage index significantly fluctuated (*p* = 0.016) with seasonal changes in the 10 y restoration period, and presented no obvious difference with more than 20 years of restoration (Figure 2b–e), which suggested that the microbial assemblages tended to be more stable in resisting seasonal fluctuations when the restoration period exceeded 20 years. However, the results indicated that the assemblage stability index was significantly related to the bacterial PD index. Srivastava et al. [55] suggested the diversity index as a powerful proxy for assessing the roles of species in ecosystem functioning, considering (phylogenetically linked) ecological differences between species, and several researchers have conducted experiments in microcosms or at global scales and have reported that microbial diversity is linked to ecosystem functioning, implying that microbial assemblages with higher diversity perform better [56,57,58,59]. Thus, microbial assemblages with higher diversity under relatively long restoration periods may have diverse functions that improve the assemblage stability of the reed rhizosphere microbiome.

Another issue should be noted. In addition to recovery time, there were likely other factors that varied with sampling locations and thus interfered with the evaluation of the effects of recovery time on assemblage stability, such as the basic conditions before recovery, and soil and water characteristics [60]. In the relatively small region (no more than 50 km in straight line distance), before discovery engineering, human activities such as agricultural production, fishery measures and wastewater discharge may be the determining factors [61,62,63]. After discovery, under little anthropogenic interference, the microbial community may evolve in the process of homogenization [5]. We studied the evolutionary mechanism of homogenization, and evaluated the functions of the microbial indicators, core taxa and keystone species in the process of homogenization.

### 4.2. Bacterial Indicators Indicate the Microbial Assemblage Stability

Monitoring the dynamics of microbial communities in different niches could identify the microbial indicators and provide insights to assess environmental variations and the status of ecosystem health [64,65,66,67]. For example, previous studies reported that several taxa (such as *Alphaproteobacteria*, *Rhodospirillales*, *Caulobacterales* and *Rhizobiales*) were the dominant indicators in river or lake sediments located at a distance from human activities and indicated a minimally polluted environment [68,69,70]. The natural prokaryotic community and microbiota serve as indicators of the health of coral reefs and soil [23,25]. The results of this study showed that the number of bacterial indicators and unique species increased during the restoration periods, but the proportion of unique species decreased (Figure 3a and Appendix A). In the process of microbial restoration, unique species were continuously generated in the bacterial assemblages; however, as the restoration period extended, the organization of the communities gradually stabilized, and the proportion of unique species declined (Appendix A). Surprisingly, the highest number of indicators appeared in the 30 y group; the number of BIs in the 40 y group was lower than that in the 20 y or 30 y group (Appendix A). A possible explanation for this result is that the bacterial community structure of the niche tended to become relatively stable after 30 years of restoration, and the stable microbial community fluctuated in the normal range under restoration for more than 20 years. However, the results indicated that fewer FIs than BIs and more than 80% of unique species in the FIs occurred in different stages. In particular, in the 10 y or 20 y group in the wet or dry season, all of the FIs were unique species (Appendix A). Previous studies indicated that rhizoplane fungi are mainly composed of symbiotrophs, saprotrophs, pathotroph–symbiotrophs, and pathotroph–saprotrophs [71,72]. Therefore, the fungi had less communication with each other due to their own characteristics. Additionally, the aforementioned results indicated that the restoration period had little effect on fungal diversity, and the different fungi may be mainly affected by the host [73,74]. Thus, we concluded that relatively long restoration times increased the indicators, and the number of BIs in the rhizosphere microbiome may be a promising option for indicating the stability of microbial assemblages.

### 4.3. The High Abundance of Core Taxa and Keystone Species Contributes to Assemblage Stability

The top four most abundant modules were found in the MON, which suggested that the core taxa differed among the recovery times (Figure 4a). Previous studies have proposed that modules reflect habitat heterogeneity, divergent selection regimes, clusters of phylogenetically closely related species, and even the key unit of species coevolution [19,75,76,77]. In this study, the number of bacteria and fungi differed markedly, and the abundance of core taxa varied among the modules (Figure 4c–e and Appendix A). For example, in M2, there were no fungi, and only bacteria established a functional microbiota. However, in the other three modules, bacteria and fungi together composed of the functional assemblages (Appendix A). However, the core taxa of M2 presented high abundance in the 30 y and 40 y restoration periods, and all of the keystone species appeared in this module (Appendix A). Previous studies indicated that the modularity of core taxa played a key role in maintaining microbial assemblage stability [9,26]. Regression analysis revealed that the core taxa of M2 contributed to enhancing assemblage stability in the relatively long restoration periods (Figure 5b). Conversely, in M8, highly abundant core taxa appeared in the 10 y restoration period, and the higher the core taxon abundance was, the less stable the microbial assemblages were (Figure 5c). Moreover, the abundance of keystone species significantly increased with recovery time. Keystone species are frequently interconnected with numerous other species, and they perform powerful functions in the whole assemblage [78,79]. There was a positive relationship between keystone species and assemblage stability (R^2^ = 0.451) (Figure 5a). Thus, in this study, the core taxa and keystone species showed higher abundances in the relatively long restoration periods and may help promote assemblage stability.

### 4.4. Functions of Core Taxa and Keystone Species in Improving Assemblage Stability

With the functional prediction of the modularity of core taxa and keystone species, all the modules of the bacterial assemblages had similar functional profiles, and several modules interacted with fungal assemblages (e.g., M1, M6 and M8), where the abundance of the functional profiles exhibited great divergence (Figure 6). For example, in M2, COG0438 (carbohydrate transport and metabolism), COG0451 (nucleoside-diphosphate-sugar epimerase), COG1595 (DNA-directed RNA polymerase specialized sigma subunit, sigma24 family), COG2204 (DNA-binding transcriptional response regulator, NtrC family, contains REC, AAA-type ATPase, and a Fis-type DNA-binding domain), COG1028 (inorganic ion transport and metabolism), COG0845 (multidrug efflux pump subunit AcrA (membrane-fusion protein)), COG0583 (DNA-binding transcriptional regulator, LysR family), COG1309 (DNA-binding transcriptional regulator, AcrR family), and COG1538 (outer membrane protein TolC) were the top 10 in relative abundance of functions in the four restoration periods (Figure 6 and Appendix A). Under relatively long restoration, only the core microbiota with bacterial functional groups can have effective functional associations and improve the stability of microbial assemblages, which is consistent with the findings of a previous study [9]. Nevertheless, the abundance of core taxa in the M8 was highest in the 10 y group. (Figure 4e). The four functional profiles (COG0438, COG1028, COG0451 and COG1595) were the same as those observed for M2, and the bacterial functions COG0535 (radical SAM superfamily enzyme, MoaA/NifB/PqqE/SkfB family), COG0596 (pimeloyl-ACP methyl ester carboxylesterase), COG2814 (predicted arabinose efflux permease, MFS family), COG0642 (signal transduction histidine kinase), COG1629 (outer membrane receptor proteins, mostly Fe transport), and COG1024 (enoyl-CoA hydratase/carnithine racemase) and fungal function animal pathogen endophyte plant pathogen wood saprotroph (APEP), undefined saprotrophs(USs) and fungal parasites (FPs) were specialized in this module (Figure 6). These results are consistent with those of other studies and suggest that saprotrophic fungi are vital decomposers of dead or withered plants and play pivotal roles in decomposing organic matter and nutrient cycling [80,81]. However, the bacterial and fungal functional groups in this module negatively affected the stability of the microbial assemblage (Figure 5). Under a relatively short recovery time, the microbial community may be active against environmental pathogens, resist the pressure of pollutant fluctuations, and fluctuate greatly due to perturbations. Therefore, with an increase in the restoration period, greater communications may occur among the core bacteria, and the modules of microbiota exhibit functional diversity to enhance assemblage stability, while the core taxa in the relatively short restoration periods may vary with environmental fluctuations and have a negative effect on assemblage stability.

## 5. Conclusions

The main purpose of the current study was to examine recovery effects on the assemblage stability of the reed rhizosphere microbiome regarding phylogenetic diversity, indicators, and core taxa as well as the functions of the core taxa and keystone species. On the one hand, we found that long-term restoration helped improve microbial assemblage stability and that bacterial assemblages are a superior option for indicating the stability of the microbial assemblage during the process of restoration. The microbial assemblage may tend to stabilize by resisting seasonal fluctuations after 20 years of restoration. Thus, as a basis for incorporating microbial management strategies into intelligent wetland systems, wetland restoration should be carried out as soon as possible, and the recovery time may be at least 20 years from the perspective of assemblage stability. On the other hand, these results indicated that the modules of core taxa and keystone species had higher abundances in the relatively long restoration periods, performing diverse functions to enhance assemblage stability. We propose that reed rhizosphere microbiota manipulations and management strategies should also be considered intelligent actions. With increasing harbor time in niches, these microbiotas could associate with others by self-organization, communication and function. These interactions make a great contribution to reinforcing the robustness of ecosystems.

## Figures and Tables

**Figure 1 microorganisms-10-00937-f001:**
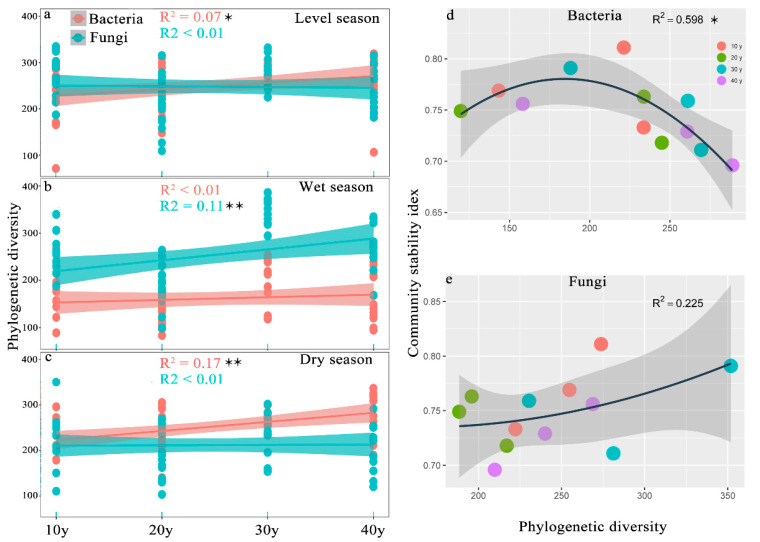
(**a**–**c**) Linear relationship between the PD index and the recovery times (significance is indicated by * *p* < 0.05, ** *p* < 0.01, *n* = 56). (**d**,**e**) Relationship between assemblage stability and the PD index. (R^2^ represents the correlation coefficient, and significance is indicated by * *p* < 0.05, *n* = 12).

**Figure 2 microorganisms-10-00937-f002:**
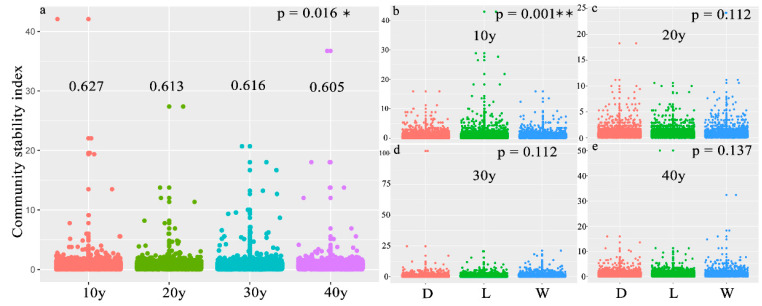
(**a**–**e**) Boxplot of the assemblage stability index in the different recovery periods or different seasons. (**a**) differenct recovery time, (**b**) 10 y, (**c**) 20 y, (**d**) 30 y, (**e**) 40 y. (The *p* value indicates the significance of ANOVA between the four recovery times or the three sampling seasons. * and ** represent *p* < 0.05 and *p* < 0.01, respectively. D: dry season, L: level season; W: wet season).

**Figure 3 microorganisms-10-00937-f003:**
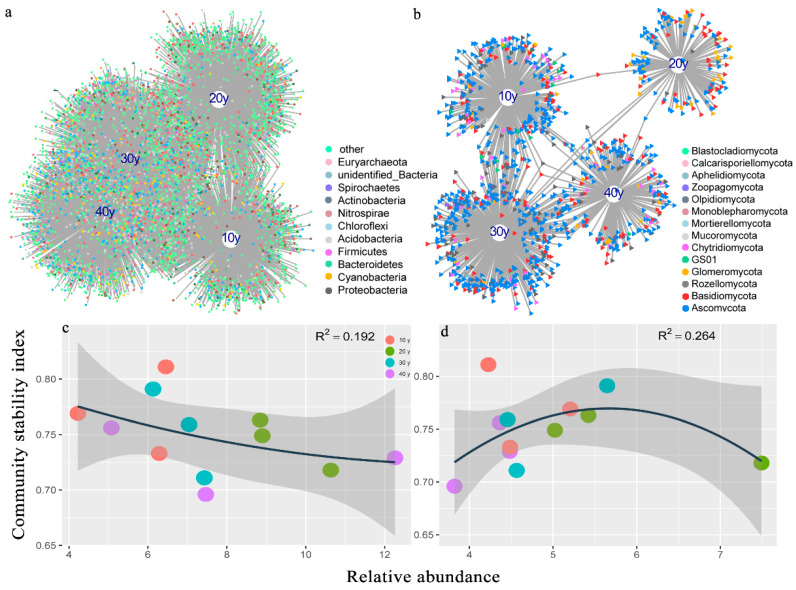
(**a**,**b**) Bipartite networks displaying specific OTU responses to recovery time in the reed rhizosphere. Bacterial and fungal OTUs were selected using indicator species analysis. Circles represent individual bacterial OTUs, and triangles represent the fungal OTUs that were significantly associated (*p* < 0.05) with recovery time (associations shown by connecting lines). OTUs are colored according to their phylum assignment. (**c**,**d**) Relationship between the assemblage stability index and the relative abundance of bacterial and fungal indicators. (R^2^ represents the correlation coefficient, *n* = 12).

**Figure 4 microorganisms-10-00937-f004:**
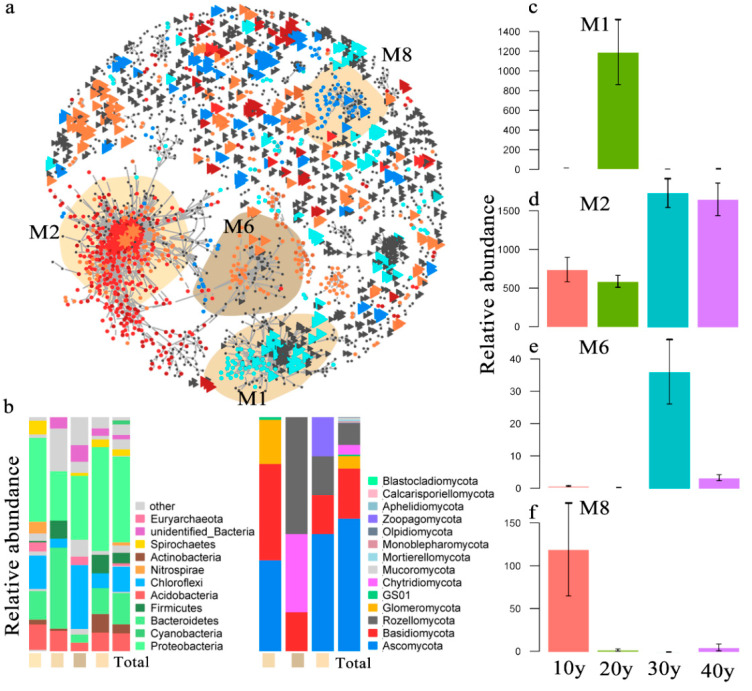
(**a**) Co-occurrence patterns of core microbiota, co-occurrence networks showing significant correlations (R > 0.7, *p* < 0.001; indicated with gray lines) between bacterial and fungal OTUs of reed rhizoplane microorganisms. Circles indicate bacteria, triangles indicate fungi, and keystone OTUs are represented with asterisks. Core taxa are colored according to their association with the different recovery times (gray OTUs were insensitive to recovery times). Shaded areas represent the network modules containing OTUs. (**b**) Qualitative taxonomic composition of core taxa in the module reported as proportional OTU numbers per phylum (bacteria and fungi) at different recovery times. (**c**–**f**) relative abundances of all bacteria and fungi of the core taxa in the module. The cumulative relative abundance in samples of different colors (firebrick (40 y related), sienna (30 y related), cyan (20 y related) and dodger blue (10 y related)) indicates the overall response of core taxa to the different recovery times.

**Figure 5 microorganisms-10-00937-f005:**
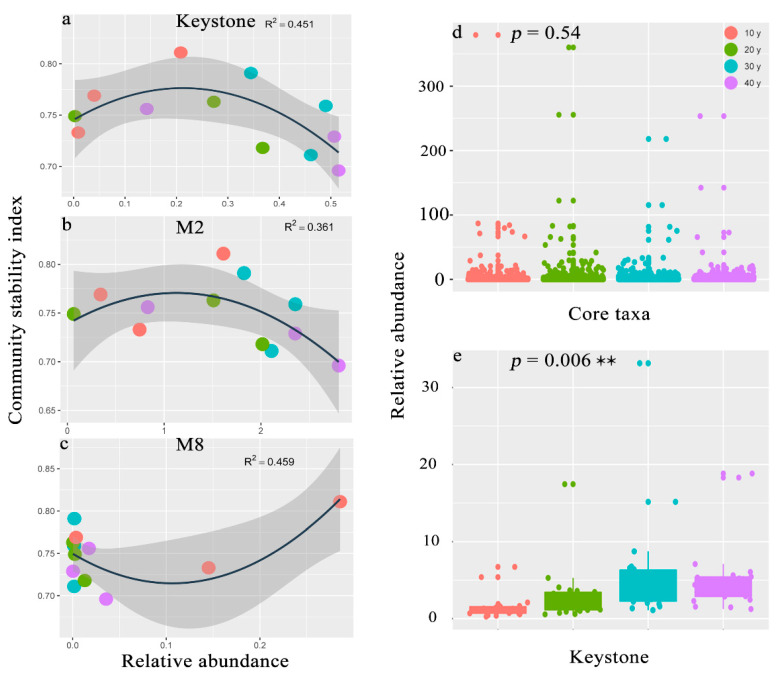
(**a**–**c**) Relationship between the assemblage stability index and the relative abundance of core taxa and keystone species (R^2^ represents the correlation coefficient). (**d**,**e**) Boxplot of the relative abundances of core taxa and keystone species in the different recovery periods (the *p* value indicates the significance of ANOVA between the four recovery times, statistically significant ** *p* < 0.01).

**Figure 6 microorganisms-10-00937-f006:**
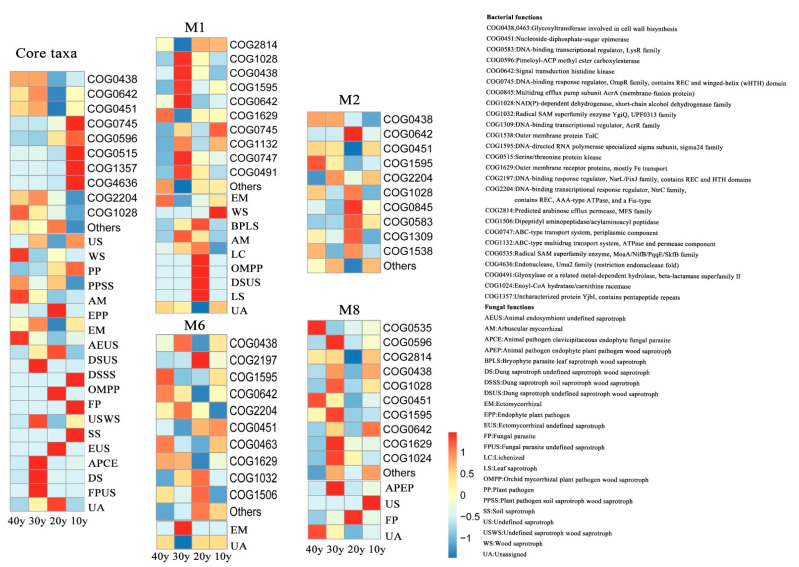
Bacterial and fungal functional predictions for the core microbiota as annotated by PICRUSt 2 and FUNGuild. (Core taxa, M1, M2, M6 and M8).

## Data Availability

Not applicable.

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
