# Peer review of "Gradual Enhancement of the Assemblage Stability of the Reed Rhizosphere Microbiome with Recovery Time"

_microorganisms, 2022, doi:10.3390/microorganisms10050937_

Round 1

Reviewer 1 Report

The main aim of the paper was to examine recovery effects on the community stability of the reed rhizosphere microbiome regarding phylogenetic diversity, indicators, and core taxa as well as the functions of core taxa and keystone species.  

The paper is well documented and the interpretation of the results is satisfactory. According to me, the reviewed paper deserves to be published in Microorganisms. I recommend providing the language changes made by native-speaker.

Author Response

Response letter

[Fuchao Zheng: School of Life Science and Institute of Wetland Ecology, Nanjing University, Nanjing, Jiangsu, China, 210093. E-mail: [email protected]]

Dear, peer reviewer

Thursday, April 21, 2022

Very pleased to contact with you!

Thank you for your valuable comments. According to your suggestions. We feel sorry for our poor writing. Our manuscript has been edited by AJE. We hope the revised manuscript could be acceptable for you. If you have any problems, please don't hesitate to contact us.

We look forward to hearing from you.

Sincerely,

Fuchao Zheng

Reviewer 2 Report

Authors described differences in reed rizosphere in several areas that undergo wetland restoration.  Examined reas were restored before 40, 30 20 and ten years. Authors used 16S rRNA gene and ITS region for analysis of microbial community. This can be considered as standard method. Authors choose to use Ion torrent sequencing instead more common Illumina one.

Introduction is written toward network analysis and its advantages. However such analysis should be companied with other widely used attitudes to draw complex picture and maintain comparability with other studies.

L107-110 Please specify total number of samples which were taken. Apparently from figures, there is sufficient number of samples, but it is slightly hard to recognize. Is there 14 transects in each season (14x3) and in each season 4 samples in each transect (14x3x4)?

Table of  all samples (not only sites) accompanied with their properties (like date of collection)  in supplement will be very helpful in this. It will be perfect if OTU table will be included. Also some additional information about wetland soils (chemical, physical properties) is important.

L126 You analysed only V4 region of 16S.

L127 Original method of Walters et al. did not mention primers ITS 1F-1R. It should be ITS1-F ITS2. Acquired data seems to be analyzed in sufficient way.

L150 I did not understand what exactly is means the statement about 30 most abundant phyla.

Results.

L184 Please add the word "gene" after 16S rRNA and remove "rRNA" after ITS because you don’t analysed RNA but DNA and ITS is not a part of rRNA. 

Figure 1 I am not sure if using quadratic regression is optimal for only 4 point. Probably not.  Also low R2 values indicate it. Thus this part of results should be removed. I also did not consider relation of PD with stability index as relevant. Overall all regression analyses in article seem weird.

Instead I would see comparison of diversity indices and stability index (ANOVA or Wilcoxon test, according normality of data). I encourage authors to move it from supplement into main article and conduct pairwise comparison in each analysis.

Moreover I did not found any comparison of community composition like ANOSIM or PERMANOVA. This should be added.

There was not any shared OTU according Venn diagram in supplement? Weird, I never saw such study (I conduct, reviewed and read large number of such studies). Is it caused by sequencing platform (Ion torrent is known for its problems in homopolymer parts), no appropriate data analysis or I did not understand what exactly is in Venn diagram?  

In figure 3, marks of phyla are very small. Again regression is not appropriate.

Figure 4 what exactly is “Cumulative relative abundances (as counts per  million, CPM; y-axis in ×1000)”? If it is a sum of relative abundance in all samples in single treatment. Please use rather relative relative abundances, simply sum relative abundances and divide them by number of samples.

Instead functional prediction based on module core taxa authors should use comparison of functional prediction of community within all samples. Then compare functional profiles among treatments (recovery periods) and seasons.

Through whole study, authors consider samples to be different only due their recovery periods, however there are presented by different sites in nature. These sites were not started from the same status, thus they can be different simply due to other factors than restoration period. This should be mentioned in discussion.

L445 What is Kim microbiota?

Data availability warning: Before publishing, all sequence based studies must have deposited sequences in public repository like Genbank. There is not any mention of data deposition in current manuscript. However I was not able to open supplementary link
Doi: 10.5281/zenodo.6340042 thus I dont know if there is such information.

Before publishing, article need undergo significant changes including new statistical analysis.

Author Response

Response letter

[Fuchao Zheng: School of Life Science and Institute of Wetland Ecology, Nanjing University, Nanjing, Jiangsu, China, 210093. E-mail: [email protected]]

Dear, peer reviewer

Thursday, April 21, 2022

Very pleased to contact with you!

Thank you for the nice comments on our text, According to your comments. We have made extensive modifications to our manuscript and supplemented extra data to make our results convincing. The comments of peer reviewer were replied point by point as follows. If you have any problems, please don't hesitate to contact us.

We look forward to hearing from you.

Sincerely,

Fuchao Zheng

Reviewer 3 Report

Review of the manuscript entitled “Gradual enhancement of the community stability of the reed rhizosphere microbiome with recovery time” by Zheng et al., submitted to Microorganisms for consideration.

General comment: With this work, the Authors want to study “the stability, diversity, abundance, co-occurrence patterns, and functional characteristics of reed rhizosphere microbes in restored wetlands”, and mainly conclude that “community stability significantly increased with recovery time, and the microbial communities were capable of resisting seasonal fluctuations after more than 20 years of restoration”.

The Authors also provide some conclusions regarding “bacterial indicators”, as well as “core taxa” and “keystone species”, “community stability” and “microbial functions”, proposing that their work would “present a natural approach for the empirical management of wetland ecosystems”.

Overall, I found this work very confusing or not convincing, due to several weaknesses and/or statements that are not sufficiently sounding in terms of scientific rigor. I provide here below some specific comments, which I hope may help the Authors to identify major weak points and improve the data analysis and interpretation.

Specific comments:

As the Authors did not perform microscope-based analyses, I suggest to avoid any statement about microbial “abundance”. In mst sentences, “abundance” should just be replaced with “OTU” or “sequence” “number” or “sequence reads” or similar, as the dataset is based on 16S rDNA reads, not cell counts.

Throughout the manuscript and in the title, I suggest the use of “assemblage” instead of “community” in case of microbial assemblages.

Overall, many sentences are unsupported by the cited references or miss reference at all. For example, REF #21 Martin et al (L60-62) did not use network analysis and did not reach such conclusions. REF #25 Philippot et al (L66-68) did not discuss core taxa nor stability in their work. The analysis on “community stability”, which is fundamental for the main conclusions of this paper, lacks proper reference and sufficient description of the method utilized (Ref #41 and #42 do not deal with “average community stability-AVD” as intended by the Authors, and also differ in their relative mathematical approach. And based on what is written, ref #14 apparently is the only previous paper using this “AVD” index, though in the context of an experimental, simplified system). L179-181 again, missing proper reference for this.

For data presented in Figure 1A,B,C. The simple linear regression is likely not the correct statistical tool here to assess significant differences. I argue that ANOVA with “time” as a factor, would reveal no statistical differences between the 4 time points, thereby changing the overall data interpretation. Similarly, for data in Figure 1D,E,F; and Fig 3 C,D; and Figure 4 A,B,C (in these cases the hypothesized relationship is evidently not conserved within each year, and in all cases a single dot -or few- drives the trend. Conclusions based on these trends are not robust.

Figure 2. It is not so clear which comparison the p values refer to. The figure shows values which appear nearly identical, while two p values (Fig.2A,B) seem to highlight statistically supported differences, which is strange.

Figure 3A,B the color legend is not visible. Moreover, the expressions “restorative effects of specific OTUs” and “OTUs that were positively and signif-249 icantly associated (p < 0.05) with one or more of the restorative effects (associations shown by con-250 necting lines)” are not clear, please better explain.

Figure 4 D, Keystone OTUs are represented with asterisks”, but I cannot see asterisks. Moreover, not clear if these should be bacterial or fungal. Please fix

Throughout the manuscript, several sentences sound a bit confusing and/or represent statements expressed with insufficient scientific rigor. Some examples at L20 “The bacterial indicators were greater in the restoration groups with longer restoration periods“; L24-26 “Our study provides insight into the connection between the rhizosphere microbiome and recovery periods and presents a natural approach for the empirical management of wetland ecosystems”; L222-224 “All of the species were used to identify bacterial and fungal indicators […] whose abundances significantly differed among the recovery times in the reed rhizosphere”; L378 “indicating that only bacterial functional groups could effectively cooperate 378 with each other”; L 379 “bacteria and fungi needed to work together for effective collaboration”; L320-323 […]; L329-332 […]; L335-337 […]; L341 this whole paragraph is particularly difficult to understand: the English syntax is maybe ok, but the terms used and the overall meaning are not scientifically/quantitatively sounding.

Figure S4. Data presented in this figure are only marginally discussed in the text and legends are difficult to see

Figure S5. It is not clear how L154-156 of the methods section can lead to Figure S5 and how these data are interpreted (the whole paragraph at L212 is quite confusing.

Overall, besides indicator taxa and co-occurrence network analysis, the Authors should complement their analysis by providing general information about core/shared OTUs, as well as including deeper discussion about the taxonomic information across samples.

Author Response

(The authors gave the same response as above.)

Reviewer 4 Report

Dear Authors, 

Thank you for interesting paper. 

Comments:

To make the manuscript more clear and understandable you can divided 2.3 Data analysis subsection into three separate paragraphs. 

Line 58: simple

Please improve the quality of Figures - at least Figure 4 should be enlarged or divided to two separate figures, but all the rest of Figures are too small. 

Line 320: plant are biomass not accumulate the biomass

Line 445: Kim microbiotas?

Discussion section should be definitely enriched with the information about relationship between particular COGs and its probable role in the mentioned wetlands, this is for sure the weakest point of your work. For example, why for M8 COGs related to Fe metabolism are present? Does it have any ecological role?

Author Response

Response letter

[Fuchao Zheng: School of Life Science and Institute of Wetland Ecology, Nanjing University, Nanjing, Jiangsu, China, 210093. E-mail: [email protected]]

Dear, peer reviewer

Thursday, April 21, 2022

Very pleased to contact with you!

Thank you for the nice comments on our text, According to your comments. We have made extensive modifications to our manuscript and supplemented extra data to make our results convincing. The comments of peer reviewers were replied point by point as follows. If you have any problems, please don't hesitate to contact us.

We look forward to hearing from you.

Sincerely,

Fuchao Zheng

Round 2

Reviewer 3 Report

The Authors have performed sufficient revisions to let me now endorse publication. Well done and best luck with your work.